# Nitric Oxide and Globin Glb1 Regulate *Fusarium oxysporum* Infection of *Arabidopsis thaliana*

**DOI:** 10.3390/antiox12071321

**Published:** 2023-06-21

**Authors:** Laura C. Terrón-Camero, Eliana Molina-Moya, M Ángeles Peláez-Vico, Luisa M. Sandalio, María C. Romero-Puertas

**Affiliations:** Department of Biochemistry, Cell and Molecular Biology of Plants, Estación Experimental del Zaidín, CSIC, Profesor Albareda 1, 18008 Granada, Spain; laura.terron@csic.es (L.C.T.-C.); eliana.molina@eez.csic.es (E.M.-M.); mpelaezvico@hotmail.com (M.Á.P.-V.); luisamaria.sandalio@eez.csic.es (L.M.S.)

**Keywords:** *Arabidopsis thaliana*, *Fusarium oxysporum*, globins, ROS, nitric oxide, PR proteins

## Abstract

Plants continuously interact with fungi, some of which, such as *Fusarium oxysporum,* are lethal, leading to reduced crop yields. Recently, nitric oxide (NO) has been found to play a regulatory role in plant responses to *F. oxysporum*, although the underlying mechanisms involved are poorly understood. In this study, we show that *Arabidopsis* mutants with altered levels of phytoglobin 1 (Glb1) have a higher survival rate than wild type (WT) after infection with *F. oxysporum*, although all the genotypes analyzed exhibited a similar fungal burden. None of the defense responses that were analyzed in *Glb1* lines, such as phenols, iron metabolism, peroxidase activity, or reactive oxygen species (ROS) production, appear to explain their higher survival rates. However, the early induction of the PR genes may be one of the reasons for the observed survival rate of *Glb1* lines infected with *F*. *oxysporum*. Furthermore, while *PR1* expression was induced in Glb1 lines very early on the response to *F. oxysporum,* this induction was not observed in WT plants.

## 1. Introduction

Plants interact with numerous microbes, leading to both negative and beneficial plant fitness outcomes. In particular, fungi play a key role in both natural and agricultural environments. Although mutualistic and neutral interactions are predominant, fungal plant pathogens cause significant losses in both greenhouse and outdoor crop production [1,2]. Specifically, two *Fusarium* species, *F. graminearum* and *F. oxysporum*, have been ranked as fourth and fifth in the top ten list of fungal plant pathogens, respectively [3]. In particular, *F. oxysporum*, which is responsible for vascular wilt and root rot disease, with its wide range of hosts, has caused severe losses in around one hundred crops, including banana and cotton [4], and also infects the model plant *Arabidopsis thaliana*. Infectious hyphae, which are able to sense signals from plants, such as the catalytic activity of secreted class III peroxidases, sugars, and amino acids, are able to enter the roots [5]. Crosstalk between the plant and the fungus is then activated until the infection develops. Recently, NO production during *Arabidopsis*/*F. oxysporum* [6] interactions have been postulated to play a regulatory role in tomato responses to *F. oxysporum* [7]. Research conducted over the last 25 years has highlighted the importance of NO as a signaling molecule in plant metabolism [8]. Since the time of the first studies, the function of NO has been closely linked to plant immunity [9,10,11,12]. NO levels have been widely observed to increase during processes, such as microbe-associated molecular patterns (MAMPs), pattern-triggered immunity (PTI), and effector-triggered immunity (ETI) responses; NO also plays an important role in hypersensitive responses (HRs) and systemic acquired resistance (SAR) [11,13,14]. Furthermore, damage-associated molecular patterns (DAMPs) have been shown to induce NO production and exhibit a feedback interaction with Ca^2+^ [15]. However, insufficient research has been devoted to the specific role played by NO during plant/pathogenic fungus interactions, especially with root fungal pathogens, probably due to the difficulties involved in analyzing below-ground activity [2,16]. These interactions appear to be determined by the lifestyle of necrotrophic, biotrophic, and hemi-biotrophic pathogens [2,17,18,19]. Thus, in the plant interactions with fungal pathogens, such as *Botrytis cinerea*, *Aspergillus nidulans*, *Macrophomina phaseolina*, *Verticillium dahlia,* and *F. oxysporum,* differential concentrations and spatio-temporal patterns of NO have been observed in the plant tissue [2,18,20,21,22,23]. On the other hand, exogenous treatment with NO has been found to reduce infection by *Rhizoctonia solani* in tomato plants [24]. 

The levels of the highly reactive molecule NO need to be tightly regulated in order to prevent unwanted damage to the cell. During the early stages of plant–microbe interactions, non-symbiotic hemoglobins/phytoglobins (Glbs) have recently been revealed to be involved in NO regulation [2,25]. Glbs are heme proteins that typically comprise a heme prosthetic group and a polypeptide composed of six to eight alpha-helix structures. The iron protoporphyrin heme is able to bind with ligands, such as diatomic gases, including O_2_, CO, and NO. While O_2_ and CO_2_ exclusively bind with ferrous iron hemes, NO can be bound by a high-affinity ferrous iron and by a low-affinity ferric iron [26]. Glb1 can regulate NO levels either through its oxidization to nitrate, or through delivery via S-nitrosylation reactions [26,27]. Thus, *Glb* expression can be strongly influenced by both biotic and abiotic stress [28]. After examining abiotic stresses, such as cold, salt, heat, and drought in rice and *Arabidopsis* plants to determine changes in phytoglobins, surprisingly, an opposite response was observed depending on the plant species [28,29,30,31]. Under biotic stress conditions, Glb1 activity was found to be associated with an increase in defense response gene expression in both cotton and *Arabidopsis* [32,33]. An *Arabidopsis* knockout mutant of Glb1 has since been found to induce an increased tolerance to *Pseudomonas* and *Botrytis* infection, accompanied with an increased expression of salicylic acid (SA), jasmonic acid (JA), and ethylene [34]. Many of the physiological changes involving Glb1, either in response to growth and development or in response to biotic and abiotic stresses, have been frequently associated with programmed cell death (PCD) [35,36]. Moreover, *Glb1* expression has been found to be activated by NO in quite a number of plant species [37,38,39]. In addition, NO activity plays an important role in transgenic lines with altered levels of Glb1 through plant–microbe interactions [34,37,39,40,41]. 

While the presence of NO and Glb1 has been previously described during plant/*F. oxysporum* interactions, little is known about the mechanisms underlying their crosstalk, or their possible functions in plant resistance to the fungus. Using a genetic approach, we analyzed the role of Glb1 in *Arabidopsis* resistance and defense responses to *F. oxysporum*. Thus, to achieve this aim, two mutants with altered Glb1 levels, including the antisense line phytoglobin 1 *35S::antiGlb1 (L3)* and the overexpressing line phytoglobin 1 *35S::Glb 1* (*H7*) [37] were assessed for *Arabidopsis* defense responses and resistance to *F. oxysporum*.

## 2. Materials and Methods

### 2.1. Plant Growth Conditions and Fungal Strains 

For the infection experiment, the pathogenic strain *Fusarium oxysporum* f. sp. *conglutinans* PHW 699-3 (ATCC 58110) [42], which is capable of infecting *Arabidopsis* [42], was used. For each assay, a microconidial suspension, previously stored in 30% glycerol at −80 °C, was freshly grown for 4 days in potato dextrose broth with glucose (20 g/L) in Erlenmeyer flasks shaken on a rotary shaker at 170 rpm and 28 °C. The spores were then isolated and quantified with the aid of a Neubauer chamber using optical microscopy [43,44]. 

*Arabidopsis thaliana* seeds (WT, Col-0), antisense lines L1 and L3 of globin 1 (Glb1; 35S::asGlb1), as well as the Glb1 overexpressing lines H3 and H7 (35S::Glb1) [37], were all surface-disinfected, stratified for 48 h at 4 °C, and then sown on Hoagland solid medium (0.5×), pH 5.6 [45]. Seeds were grown at 22 °C, irradiance of 100 µE, 60–65% relative humidity, and under 16/8 light/dark conditions for 14 days. 

### 2.2. Plant Infection Assays

*A. thaliana* wild type (WT) and mutant lines *L1* and *L3* (*35S::asGlb1*), as well as *H3* and *H7* (*35S::Glb1*) root infection assays were performed as described elsewhere [1]. Two-week-old *Arabidopsis* roots were immersed for 30 min in a microconidial suspension of 5 × 10^6^ spores/ml^−1^ of the *F. oxysporum* f. sp. *conglutinans* 699 isolate. At least sixty seedlings per treatment and genotype were planted in mini pots with soil/vermiculite (1:1) after infection. *Arabidopsis* plants were then grown in a growth chamber at 24 °C, 120 µE irradiance, 60–65% relative humidity, and under 16/8 light/dark conditions for 25 days. The plant survival rate, quantified by the proportion of dead/alive plants was measured daily using the Kaplan–Meier method, and the rates of the different groups were compared using the log-rank test as described elsewhere [1,46]. Plants with no green tissues were considered to be dead. 

### 2.3. Gene Expression and Fungal Burden Quantification Using Real-Time Quantitative PCR (RT-qPCR)

RNA was isolated using Trizol reagent (Invitrogen) according to the manufacturer’s protocol. Following this, the Ambion DNA-free DNase treatment was then applied. RNA (0.5 µg) was used to reverse the transcription process with 5× primer script RT master mix (Takara), as described elsewhere [47]. Specific primers for genes (Appendix A) were used to analyze transcript levels with the aid of the iCycler iQ5 (Bio-Rad, Hercules, CA, USA) and TB Green Premix Ex Taq (Takara), according to the MIQUE guidelines (Appendix A) [48]. Amplification efficiency was calculated using the formula E = [10 (1/a) − 1] × 100, where “a” denotes the slope of the standard curve. The relative expression of each gene was normalized to that of *TUB4*, whose stability is shown in Appendix A, and the results were analyzed following the Pfaffl method [49]. 

To quantify fungal burden, total DNA was extracted from infected roots at 0, 2, and 7 days post-infection (dpi), respectively. qPCR was performed with the *F. oxysporum*/*Arabidopsis*-specific primer *act2/TUB4.* The comparative threshold cycle (ΔΔCt) from their constitutive genes (*act2*/*TUB4*) was used to calculate the relative amounts of fungal DNA with respect to plant DNA isolation [46,50].

### 2.4. Nitric Oxide and ROS Detection

To detect total cellular reactive oxygen species (ROS) and nitric oxide (NO), seedlings were incubated with 25 µM 2′7′-dichlorofluorescein diacetate (DCF-DA) and 10 μM 4,5-diaminofluorescein diacetate (DAF-2 DA), respectively [51]. Root fluorescence was examined under a confocal laser scanning microscope (Leica TCS). The specificity of the reaction was checked by pre-incubating samples with ascorbate (Asc; 1 mM), as the ROS, and free radical scavenger or cPTIO (500 µM), as the NO scavenger, as described elsewhere [52]. Fluorescence intensity in seedling roots was quantified using Image J Fiji software [51]. Briefly, the average intensity per pixel of 3–5 independent squares per root of similar size was obtained as being the intensity of a root. The mean of intensity of a minimum of ten roots per time and per genotype was obtained and subsequently displayed.

### 2.5. Ferric-Chelate Reductase, Peroxidase Activities, and Western Blot

To measure ferric chelate reductase activity, *Arabidopsis* seedlings not infected (−) or infected (+) with *Fusarium* at 1, 3, and 24 hpi were transferred to plates containing 0.8% water Noble agar supplemented with 0.5 mM CaSO_4_, 0.5 mM ferrozine, and 0.5 mM EDTA-Fe (III). After a 20 min incubation at room temperature, the plates containing seedlings were imaged [53,54]. In addition, peroxidase activity was observed in *Arabidopsis* seedlings infected (+) or not (−) with *Fusarium* (1, 3 and 24 hpi). The seedlings were transferred to plates containing 0.8% water Noble agar supplemented with 0.91 mM ABTS and 2.5 mM H_2_O_2_. The plants were incubated for 45 min at 28 °C and then imaged [5]. Proteins from the roots, not infected (C) or infected with *Fusarium* at 3, 24, and 48 hpi were homogenized with liquid nitrogen in 50 mM Tris-HCl buffer, pH 7.0, containing 0.2% Triton X-100 (*v*/*v*), 0.1 mM EDTA, and a cocktail of protease inhibitors (Sigma). Samples were centrifuged at 13,000 rpm for 30 min at 4 °C, and the supernatants were then collected. The supernatants were quantified and prepared in 0.063 M Tris-HCl buffer, pH 6.8, containing 2% sodium dodecyl sulfate (SDS; *w*/*v*), 10% glycerol (*v*/*v*), 0.006% bromophenol blue (*w*/*v*), and 10 mM DTT, and were subsequently heated at 95 °C for 5 min. The samples were then used for electrophoresis by SDS-PAGE. Proteins contained in gels were transferred to a Millipore polyvinyl difluoride (PVDF) membrane, using a semi-dry transfer system (Bio-Rad) in 10 mM CAPS buffer (3-(cyclohexylamino)-1-propanesulfonic acid), pH 11, containing 10% methanol (*v*/*v*) at 1.5 mA per cm² for a period of 1 h. The membrane was stained with Ponceau red to check for protein loading. To detect Glb1, polyclonal anti-Glb1 was used as described elsewhere [37]. Membranes were incubated with the ECL Plus Western Blotting Detection System (Amersham-TM), according to the company’s instructions. Fluorescence was detected using a ChemiDoc detection system (Bio-Rad).

### 2.6. Quantification of Phenolic Compounds from Root Exudates

Phenolic compounds were quantified by measuring root exudates at 365 nm, as described elsewhere [55,56]. *Arabidopsis* plants, inoculated or not with *F. oxysporum* for 3 h and 24 h, respectively, were transferred to a 96-well microplate with 140 μL of distillate water per well. A Varioskan LUX multimode microplate reader was then used to detect the fluorescence emission of a 100 μL aliquot of root exudate solution (excitation at 360 nm; emission at 528 nm). 

### 2.7. Principal Component Analysis

To explore all the variables studied and to identify their patterns and interrelationships, we performed a short (0–6 hpi) and long (48–96 hpi) time principal component analysis (PCA). The long-time PCA included data regarding fungal burden, NO and ROS production, Glb1 content, defense response gene expression, and iron metabolism. In the short-time PCA, we added data regarding phenol exudates and peroxidase activity.

We used R software version 4.1.0, along with other packages, including Tidyverse v.1.3.1, FactoMineR v.2.4, Factoextra v.1.0.7, and ggpubr to handle data manipulation and visualization, to perform principal component analysis, to visualize the results of the analysis, and to customize the visualization, respectively.

### 2.8. Statistical Analyzes

Mean values in the quantitative experiments described above were obtained from at least three independent experiments, with at least three independent replicates for each experiment. Statistical analyzes were performed using either a one- or two-way ANOVA test when necessary with the aid of GraphPad Prism 6 software. Mean values for the different genotypes were compared using the Tukey’s multiple comparison test (*p* < 0.05) following two-way ANOVA analysis. The Dunnett’s multiple comparison test (*p* < 0.05) after two-way ANOVA analysis, or the Student’s *t*-test after one-way ANOVA analysis were used to compare the mean values for the different treatments in a genotype. Error bars representing standard error (SEM) are shown in the figures. 

## 3. Results and Discussion

### 3.1. NO Production and Glb1 in Arabidopsis after Inoculation with Fusarium oxysporum

Nitric oxide is involved in plant responses to different microorganisms, in particular to pathogenic fungi, although its regulation is still under investigation. NO production and its specific role in plant responses to pathogenic fungi appear to be related to plant colonization strategies which induce a precise pattern of NO accumulation [2]. Thus, we aimed to investigate the function of phytoglobin1 (Glb1) and NO regulation during the infection of *Arabidopsis* plants with *F. oxysporum*. Initially, we monitored NO production over time (0–72 hpi) using the fluorescent dye DAF-2 in both WT and Glb1-related mutants roots. We observed a significant 1.7–1.8-fold increase in NO levels in WT and *L3* plants infected with *F. oxysporum* from the initial stage of the infection up to 24 hpi, which was followed by a sharp decrease in NO at 48 and 72 hpi, respectively (Figure 1A). This is consistent with previous reports which revealed a peak in NO at the onset of *Arabidopsis* infection with *F. oxysporum* [6] and other plant-root fungal interactions, such as olive-*Verticilium dahliae* [57] and tomato-*Rizoctonia solani* [24]. Different fungal elicitors also induced an increase in NO levels [2,58,59]. Oscillations in NO levels were also observed in tomato roots at the early stages of *F. oxysporum* infection [7].

From 48 to 72 hpi, the mutants affected in Glb1 were found to have behaved similarly to those in WT associated with NO production. However, at an early time point (6 hpi), the over-expressor line *H7* showed a significant decrease in NO production (Figure 1A), suggesting that Glb1 can regulate NO levels after infection. To obtain a deeper insight into the role of Glb1 in *Arabidopsis*–*F. oxysporum* interactions, we analyzed the regulation of this gene in *Arabidopsis* roots during the early stages of the infection. We observed an induction of *Glb1* transcription in WT at 6 hpi, after which its expression fluctuated, though not significantly (Figure 1B). Similar oscillations in the presence of the protein were observed by Western blot analysis in WT, and while the protein was found to be absent in *L3* lines, it was always present in *H7* lines, even under control conditions (Figure 1C; Appendix A), as described elsewhere [37]. Although Glb1 was always detected in *H7* mutants, its induction was also observed after infection (Figure 1C; Appendix A). Variations in Glb1 may be related to the changes observed in the NO levels, suggesting that Glb1 plays a key role in NO metabolism during the early stages of the *Arabidopsis*–*F. oxysporum* interaction. Similar variations in *Glb1* expression were observed in tomato roots after infection with *F. oxysporum* and *Phytophthora parasitica,* and with the foliar pathogenic fungus *Botrytis cinerea* [2]. 

### 3.2. Plant Survival in the WT and Glb1 Lines after Inoculation with Fusarium oxysporum

In order to assess whether a change in NO and different levels of Glb1 affect fungal virulence, the roots of *Arabidopsis* plants (WT, *L3*, and *H7*, respectively) were inoculated with *F. oxysporum,* and plant survival was analyzed over the course of 20 dpi, when all WT plants were found to have died (Figure 2A,B). Representative images of plants, inoculated or not with *F. oxysporum* at 9 and 20 dpi, respectively, are shown in Figure 2A. Surprisingly, *H7* and *L3* lines showed higher survival rates with respect to the WT at around 40 and 60%, respectively (Figure 2A,B). This result was independent of the amount of fungus inside the plants, which was similar across the three genotypes as a result of the fungal burden analyzed at 2 and 7 dpi, respectively (Figure 2C). We obtained similar results with the overexpressor *H3* and antisense *L1* lines (Appendix A), as described elsewhere [37]. 

In the Tomato–*F. oxysporum* interaction, fungal chemotropic growth in the roots is mediated by root peroxidases [5]. To further determine whether *Glb1* lines exhibit differential chemo-attraction to *F. oxysporum* with respect to WT, we analyzed the peroxidase activity exuded by *Arabidopsis* roots into the adjacent medium. We observed an increase in peroxidase activity in the WT roots at 1 hpi, an activity which, in the *L3* line, was similar to that of the WT (Figure 3A,B). Interestingly, *H7* mutants showed significantly lower peroxidase activities in the roots following *F. oxysporum* inoculation as compared to the WT (Figure 3A,B). No significant differences in peroxidase activity were observed between the infected plants and those not infected with *Fusarium* at 3 and 24 hpi, respectively (Appendix A). Differences in peroxidase activity between both *Glb1* lines at 1 hpi suggest that peroxidase-dependent chemo-attraction to the fungus may not be involved in the higher survival rate that was observed in both *Glb1* lines.

The different resistance phenotypes revealed that plants with altered Glb1 levels depend on the species and pathogen. Thus, the overproduction of alfalfa *Glb1* in tobacco plants reflected reduced cell death in response to either the tobacco necrosis virus, or to the pathogenic bacterium *Pseudomonas syringae* [60]. In the *L3* and *H7* lines, no effect on the hypersensitive responses (HRs) was observed during the incompatible interaction elicited by the *P. syringae* bacteria carrying the avirulence factor *Rpm1* (*Pst AvrRpm1*) [37]. On the other hand, a silenced *Arabidopsis* line, with 2–3% *Glb1* expression with respect to the WT, exhibited an enhanced resistance to *Pst* and *Pst AvrRpm1*, while *Glb1* overexpressor lines showed the opposite phenotype [34]. Furthermore, although the *35S-Glb1* line was more susceptible, and the *Glb1* gene was more resistant to the necrotrophic fungus *B. cinerea*, both lines showed significantly more electrolyte leakage than the WT [34]. Similarly, *Glb1* overexpression in barley compromised basal resistance against pathogens [61], while the overexpression of the soybean globin (GmGlb1-1) gene was found to reduce plant susceptibility to the nematode *Meloidogyne incognita* [62]. Recently, tomato *RNAi-Glb1* lines have been shown to have a more susceptible phenotype against *F. oxysporum,* while overexpressing plants exhibited a more resistant phenotype than the WT [7]. Spatio-temporal NO production and its regulation by Glb1 are both important in defining the role of NO, and altered *Glb1* expression levels may exhibit different effects depending on the specific time point in the infection process, which sometimes leads to a similar phenotype of either overexpression or silencing of the protein. Thus, an increased mycorrhizal colonization of tomato roots was observed in both silenced and overexpressed *Glb1* lines [7], while overexpression or silencing of Glb1 from *Medicago* species accounts for 30% of nodule establishment [25].

### 3.3. Iron Metabolism in Arabidopsis Plants in Response to F. oxysporum

The regulation of iron homeostasis is one of the main techniques used to control host pathogen interactions given that plants use scavenging strategies to decrease pathogen accessibility and virulence [63]. *Arabidopsis* uses the root-specific strategy I, which increases Fe uptake when necessary [64,65]. Initially, the soil is acidified by H^+^-ATPases localized in the plasma membrane in order to enhance the solubility of Fe [66]. The transcription factor FER-like iron deficiency (FIT), a master regulator of this strategy, regulates the expression of different Fe uptake genes, such as that coding for the enzyme ferric reduction oxidase 2 (FRO2) and the high-affinity iron regulated transporter 1 (IRT1), which reduces soluble Fe^3+^ to Fe^2+^ and transports Fe^2+^ to the plant root, respectively [66]. We therefore analyzed *IRT1* and *FIT* expression, as well as ferric chelate reductase (FCR) activity related to iron metabolism in *Arabidopsis*–*F. oxysporum* interactions. No significant changes were observed in either *IRT1* or *FIT* expression relative to *TUB4* expression after *F. oxysporum* infection, although *IRT1* expression relative to *TUB4* expression was found to be significantly higher in WT at 48 hpi with respect to both *Glb1* lines (Figure 4A,B). Only after 96 hpi was FIT expression relative to TUB4 expression significantly induced in the *L3* line. Although we detected ferric chelate reductase activity after 1 h of *Fusarium* infection, no significant differences were observed in the genotypes or with respect to non-infected roots (Figure 4C). We did not detect any ferric chelate reductase activity at 3 or 24 hpi, respectively (Appendix A). It has been reported that the NO scavenger cPTIO inhibits the induction of *FIT1*, *FRO2,* and *IRT1,* and that the presence of NO inhibits FIT1 degradation [67]. Furthermore, exogenous applications of ethylene and NO induce *FRO2* and *IRT1* in *Arabidopsis* plants, thus enhancing iron uptake [68]. In our experiments, however, no induction of strategy I was observed, at least during the first 96 hpi, and no significant differences were detected in either of the *Glb1* lines with respect to the WT plants. Crosstalk between different hormones also appears to be involved in the regulation of Fe-dependent genes, as salicylic acid (SA) induces *FRO2* and *IRT1* [69], and jasmonic acid (JA) inhibits their induction independently of *FIT1* [70]. Both *Glb1* lines behave in a similar way to Fe-dependent genes, suggesting that this behavior may not be associated with the higher resistance to *F. oxysporum* observed in both lines.

### 3.4. Response of Phenols and Reactive Oxygen Species in Arabidopsis to F. oxysporum

We next determined how the fungus induces defense responses during plant infection in our experimental system. Phenols, which are compounds produced via the phenylpropanoid pathway, have been shown to be involved in cell wall lignification that inhibits fungus penetration, and may also act as immunity-inducing antimicrobial molecules for the host plant [24,56]. Kinetic analysis of phenolic compounds from root exudates during *Fusarium*–barley root interactions have detected the production of t-cinnamic, p-coumaric, ferulic, syringic, and vanillic acids after day two of fungal inoculation. All these compounds after *Fusarium*–barley interactions were found to inhibit *Fusarium* spore germination [71]. We analyzed phenol production at 3 and 24 hpi, respectively, and observed that at an early time point (3 hpi), phenolic exudates decreased in response to the fungus in both the WT and mutants. Conversely, *F. oxysporum* triggered phenol production in WT root exudates at 24 hpi (Figure 5A), which coincided with the peak NO levels, although no induction of phenols was observed in any of the *Glb1* lines (Figure 5A). Although data regarding the connection between phenol synthesis and NO are scarce, exogenous NO has recently been shown to increase phenol content in tomato–*Rhizoctonia solani* interactions independently of the susceptibility of the cultivar used [24]. Interestingly, phenol levels in both mutants, *H7* and *L3*, were half of those observed in the WT, thus suggesting that Glb1/NO levels need to be optimal for phenol production to occur.

Early plant defense responses to root pathogens also include ROS production, with H_2_O_2_ being the most stable species that is directly involved in the reinforcement and cross-linking of the cell walls and defenses [72]. We detected a transient burst of ROS in WT roots at 6 hpi with *F. oxysporum* (Figure 5B). After this initial peak, ROS production varied over time, and peaked again at 48 hpi. ROS levels were always significantly higher in the inoculated WT roots compared to the non-inoculated roots (Figure 5B). *H7* lines followed the same trend as WT after *F. oxysporum* inoculation, although ROS levels in *H7* roots were 50% lower than in WT after infection, except at 72 hpi, when differences in the ROS levels between *H7* and WT were not significant (Figure 5B). On the other hand, *L3* lines showed a continuous decrease in ROS levels after the initial peak, which was similar to the decrease observed in the WT (Figure 5B). An early increase in H_2_O_2_ production after *F. oxysporum* inoculation and *F. oxysporum*-derived elicitor treatment has also been described in *Arabidopsis* roots and *T. chinensis* culture cells, respectively [6,58]. In *T. chinensis* culture cells, a decrease in the peak levels of H_2_O_2_ was observed following the introduction of an exogenous scavenger or inhibitor of NO [58]. In pea guard cells, chitosan-induced NO production occurs downstream of ROS [59]. Interestingly, the peak levels of NO and ROS differed in our experiments, with NO peaking at 24 hpi and ROS peaking at 6 and 48 hpi, respectively. Globins may interfere with the redox state of the cell, particularly with the ascorbate-glutathione cycle regulating enzymes, such as monodehydroascorbate reductase and ascorbate peroxidase [35,73]. Various studies of *Glb1* overexpression have also highlighted altered ROS levels. Recently, GmGlb1-1 has been shown to affect the dynamics of ROS production and NO scavenging, which enhances the acquired systemic acclimation to biotic and abiotic stresses [62]. Furthermore, in overexpressing *Lotus japonicus* lines, increased levels of Glb1 protect nodule symbiosis under flooding conditions by controlling ROS levels and scavenging NO [74]. *NtGlb1* expression reduces Cd levels by regulating Cd transporter expression via decreased NO and ROS levels in *Arabidopsis* [75].

### 3.5. Arabidopsis Defense Response to F. oxysporum

We further explored whether alterations in Glb1 could be related to changes in the plant defenses associated with altered NO levels. The transcript levels of *Arabidopsis* immunity marker genes, including *PDF1.2*, *PR-1*, and *PR-5* [76,77] were analyzed. We observed an increase in the expression of the JA-related defense protein *PDF1.2* in the WT at 1 hpi, which is similar to results previously reported for *Arabidopsis* [76,77]. This induction is significantly different from that observed for the *Glb1* lines (Figure 6A). Interestingly, both *Glb1* lines, *H7* and *L3,* exhibited a strong induction of the NO/SA-dependent genes, *PR1* and *PR5*, at an early time point (3 hpi) in response to *F. oxysporum,* while no corresponding significant induction was observed in WT plants (Figure 6B,C). In addition, under control conditions, *PR1* expression was significantly higher in both *Glb1* lines than in the WT (Figure 6B), thus suggesting that this defense gene is involved in the survival of *H7* and *L3* lines after *F. oxysporum* infection. Different pathogenesis-related proteins were overrepresented in the proteome of barley over-expressing globins under control conditions and after *Blumeria graminis* infection [78]. Higher basal levels of defense genes were also observed in the *Glb1* over-expressor line of the tomato [7], suggesting that Glb1 plays a role in regulating defense responses, an issue which requires further research. Recently, an ethylene-dependent increase in *Glb1* under hypoxia was shown to promote the ERFVII group’s transcription factors by limiting their NO-dependent proteolysis through the PRT6 N-degron pathway. This activates the transcription of ERFVII gene targets [79] and shows that Glb1 can play a role in gene regulation that could affect the defense genes.

### 3.6. Principal Component Analysis

A PCA was performed to examine the behavior of the different parameters analyzed after *Fusarium* infection of the *Arabidopsis* WT, *H7,* and *L3* lines at short (S) and long (L) time scales. The PCA score plots show the distribution of our experimental analyzes according to two principal components that accounted for more than half of the 56–64% variance in the 11/9 variables tested at short/long time scales (Figure 7). In addition, the PCA results show that the WT and *Glb1*-infected lines were completely separate from each other, which may be due to early defense gene induction. This can be observed in the biplot graph in Figure 7A, where both mutants infected at the short time scale are distributed according to the first principal component (PC1). At the long time scale, however, the parameters PR1 and FIT appear to be key parameters in the *H7* and *L3* line responses, respectively (Figure 7B).

## 4. Conclusions

We demonstrated that fine-tuned NO accumulation is required for proper plant responses to *Fusarium oxysporum* infection. Our results show that Glb1 is able to control the levels of NO during *A. thaliana–Fusarium oxysporum* interactions, for which transcriptomic Glb1 regulation is essential for. Interestingly, the different *Arabidopsis* Glb1 lines (both antisense and overexpressor) exhibited a more resistant phenotype than the WT in response to *F. oxysporum,* which was probably due to an early enhancement of defense gene expression. These results suggest that Glb1 may play a role in the regulation of defense genes, probably via the PRT6 N-degron pathway.

## Figures and Tables

**Figure 1 antioxidants-12-01321-f001:**
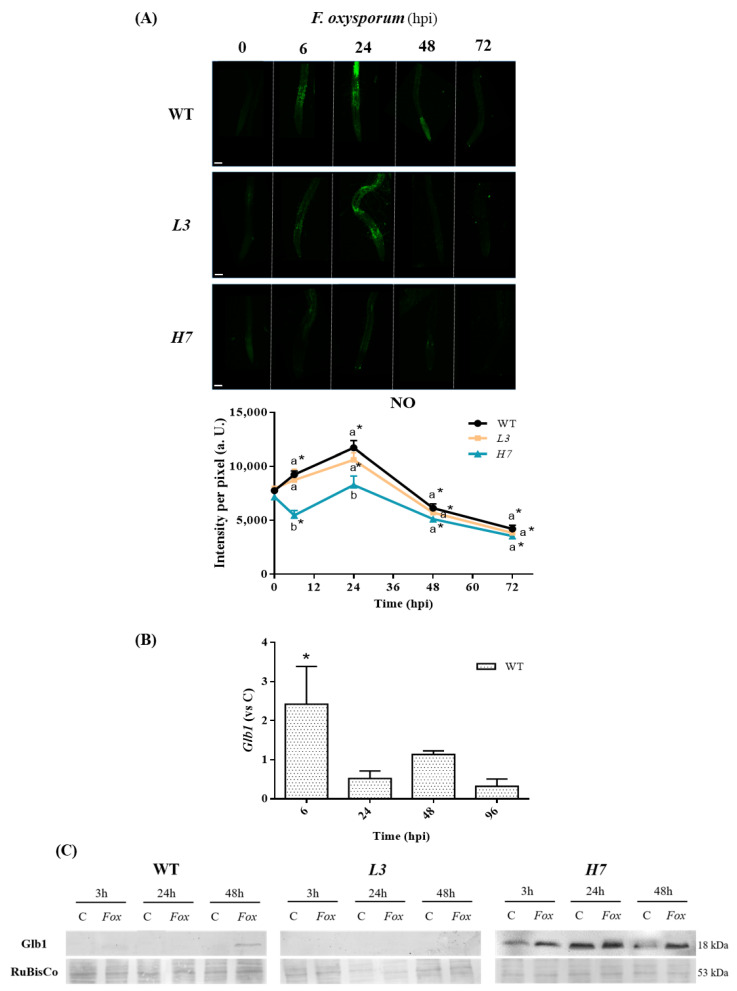
NO and Glb1 levels in *Arabidopsis* roots in response to *F. oxysporum*. (**A**) Representative confocal microscopy images of NO-dependent fluorescence using the fluorescent probe DAF-2DA in WT, *L3,* and *H7 Arabidopsis* seedling roots previously inoculated or not (0 hpi) with *F. oxysporum* (6–72 hpi), as well as image quantification. Scale bar = 100 µm. (**B**) RT-qPCR analysis of *Glb1* expression levels in WT seedlings treated with *F. oxysporum* (6–96 hpi) relative to control values at the different time points. (**C**) Representative Western blot with protein content of Glb1 in WT, *L3*, and *H7 Arabidopsis* seedling roots under control and *F. oxysporum* treatment conditions (*Fox*; 3, 24, and 48 hpi, respectively). Protein content, detected after Red Ponceau staining, is also shown for reference purposes. Quantification of two independent Western blots is shown in Appendix A. The data in A and B represent the mean ± SEM of at least three independent experiments. Different letters denote significant differences between the genotypes at each time point according to the Tukey´s multiple comparison test (*p* < 0.05). Asterisks denote significant differences with respect to the control (0 hpi) in a genotype according to the Dunnett´s multiple comparison test in (**A**) and the Student’s *t*-test (*p* < 0.05) in (**B**).

**Figure 2 antioxidants-12-01321-f002:**
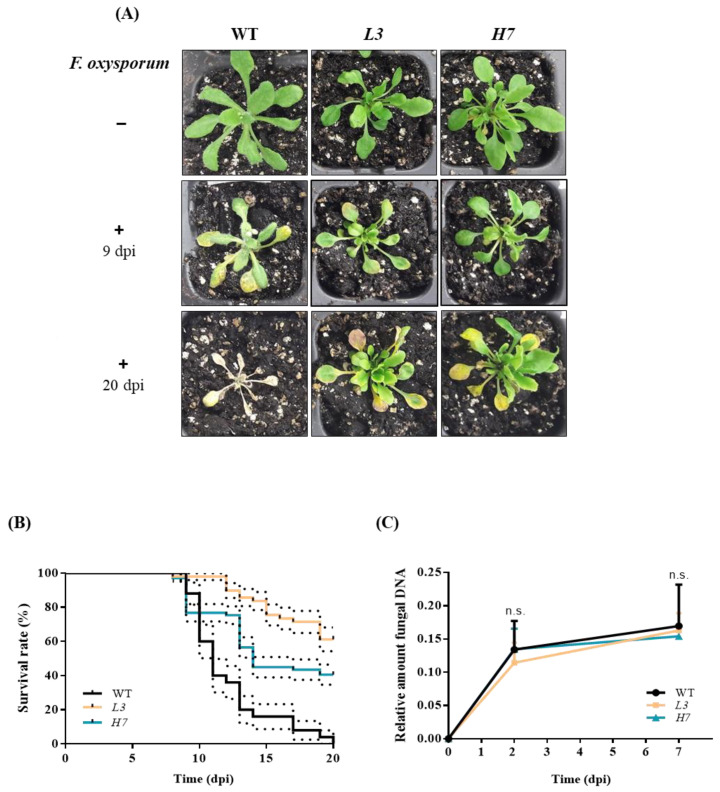
*Arabidopsis* survival after *F. oxysporum* infection and fungal burden. (**A**) Phenotype showed by WT, *L3,* and *H7 Arabidopsis* plants after 9 and 20 dpi infected (+) or not (−) with *F. oxysporum*. Scale bar = 1 cm. (**B**) Kaplan–Meier plot of the different genotypes showing *Arabidopsis* survival after infection with *F. oxysporum* over the course of 20 dpi. (**C**) Fungal burden at 0, 2, and 7 dpi determined by RT-qPCR analysis of the *F. oxysporum actin* gene relative to the *Arabidopsis TUB4* gene. Data represent the mean ± SEM of at least three independent experiments. There were no significant differences in (**C**) between the genotypes at any of the time points according to the Tukey´s multiple comparison test (*p* < 0.05; n.s.).

**Figure 3 antioxidants-12-01321-f003:**
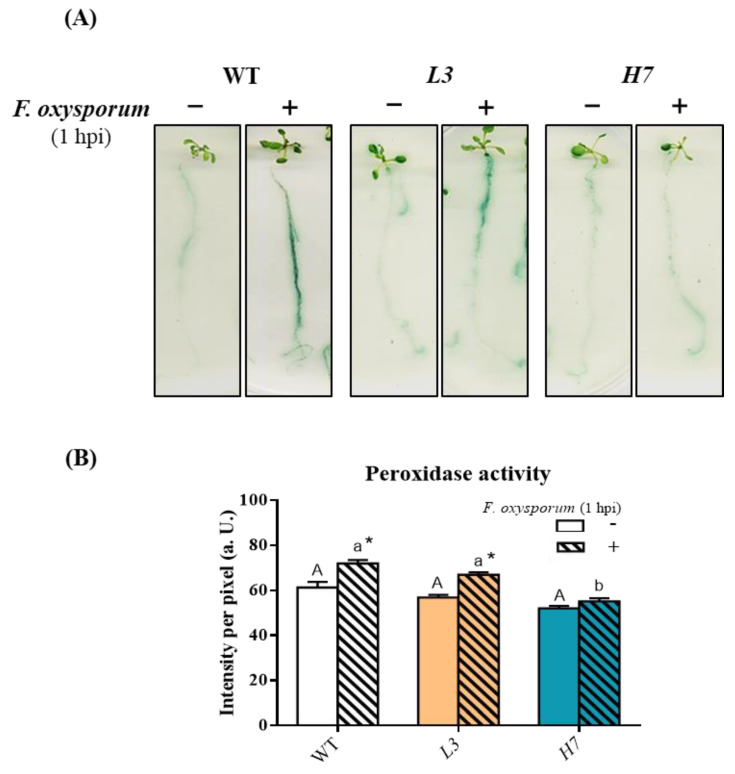
Peroxidase activity in *Arabidopsis* roots after *F. oxysporum* infection. (**A**) Representative images showing peroxidase activity in WT, *L3,* and *H7 Arabidopsis* seedling roots before (−) and after (+) *F. oxysporum* inoculation (1 hpi) and (**B**) image quantification. Data represent the mean ± SEM of at least three independent experiments (number of seedlings per experiment and genotype in (**B**) = 12). Different letters denote significant differences between the genotypes according to the Tukey´s multiple comparison test (*p* < 0.05; capital letters for control and lowercase for infected roots). Asterisks denote significant differences with respect to the control within each genotype according to the Dunnett´s multiple comparison test (*p* < 0.05).

**Figure 4 antioxidants-12-01321-f004:**
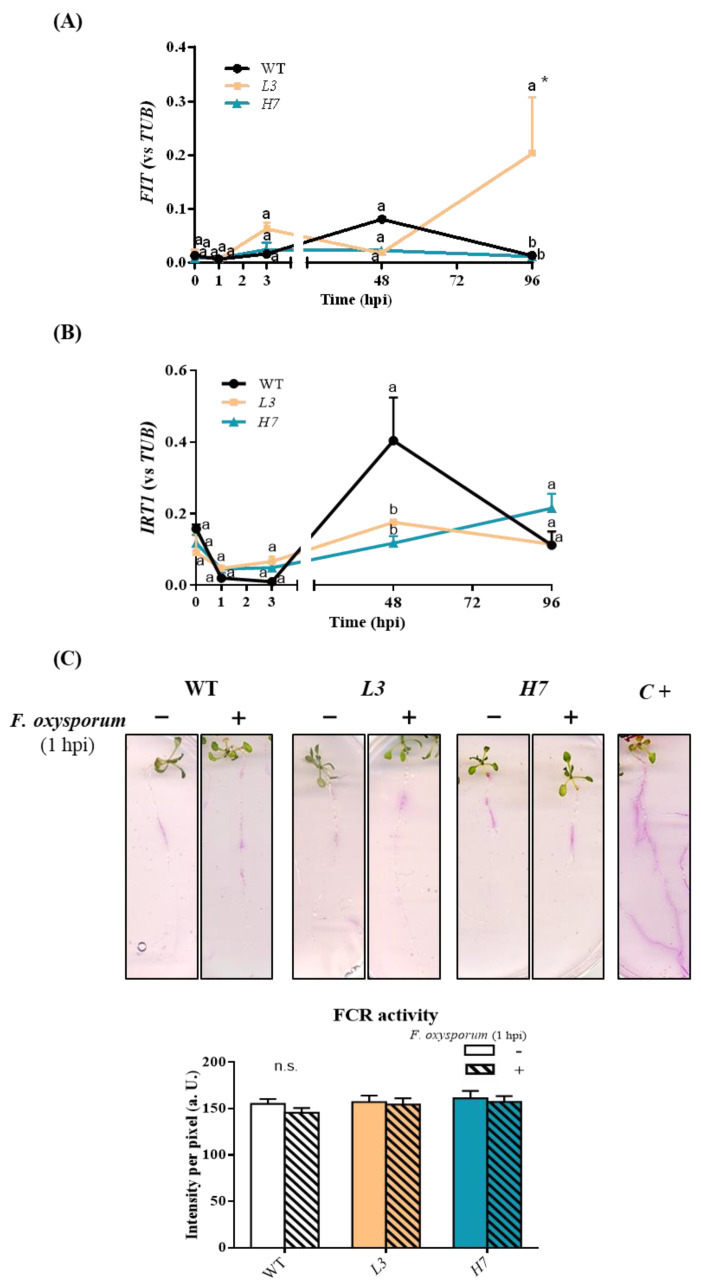
Effect of *F. oxysporum* on iron metabolism in *Arabidopsis* seedlings. RT-qPCR analysis of *FIT* (**A**) and *IRT1* (**B**) expression levels in WT, *L3,* and *H7 Arabidopsis* seedlings after *F. oxysporum* inoculation (0–96 hpi). (**C**) Representative images showing ferric chelate reductase (FCR) activity of the different *Arabidopsis* seedling root genotypes before (−) and after (+) *F. oxysporum* inoculation (1 hpi), as well as image quantification. Data represent the mean ± SEM of at least three independent experiments (number of seedlings per experiment and genotype in (**C**) = 12). Time 0 hpi refers to the mean of the control values at the different time points. Different letters denote significant differences between the genotypes at each time point according to the Tukey´s multiple comparison test (*p* < 0.05). Asterisks denote significant differences with respect to control (0 hpi) according to the Dunnett´s multiple comparison test (*p* < 0.05).

**Figure 5 antioxidants-12-01321-f005:**
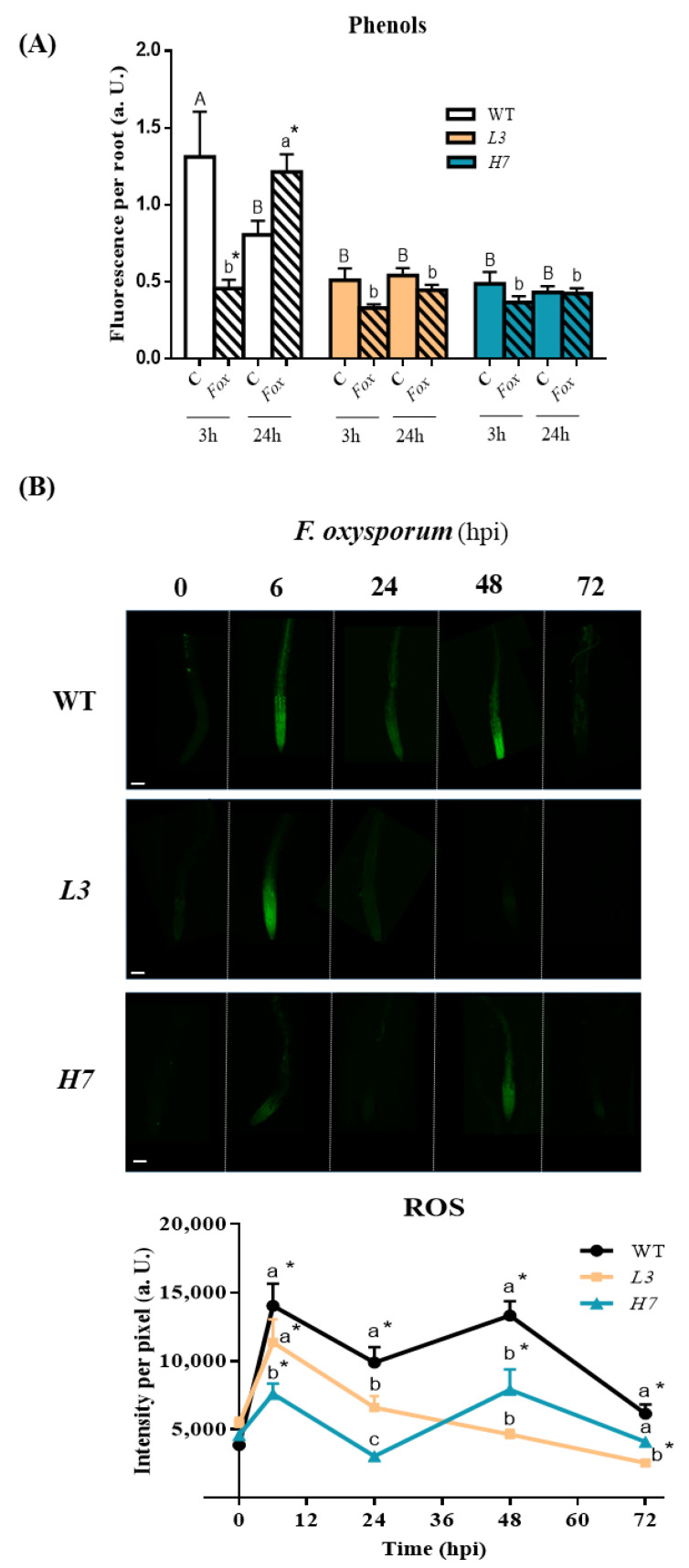
ROS and phenol accumulation in *Arabidopsis* roots in response to *F. oxysporum*. (**A**) Total amount of exudate phenolic compounds from WT, *L3,* and *H7 Arabidopsis* seedling roots infected (+; 3 and 24 hpi) or not (−) with *F. oxysporum*, determined by fluorimetry under UV light (365 nm). (**B**) Representative confocal microscopy images of ROS-dependent fluorescence using the fluorescent probe DCF-DA in *Arabidopsis* seedling roots from the different genotypes previously inoculated (6–72 hpi) or not (0 hpi) with *F. oxysporum*, as well as image quantification. Scale bar = 100 µm. Data represent the mean ± SEM of at least three independent experiments. Different letters denote significant differences between the genotypes at each time point according to the Tukey’s multiple comparison test (*p* < 0.05; capital letters for control and lowercase for infected roots). Asterisks denote significant differences with respect to the control (0 hpi) according to the Dunnett´s multiple comparison test (*p* < 0.05).

**Figure 6 antioxidants-12-01321-f006:**
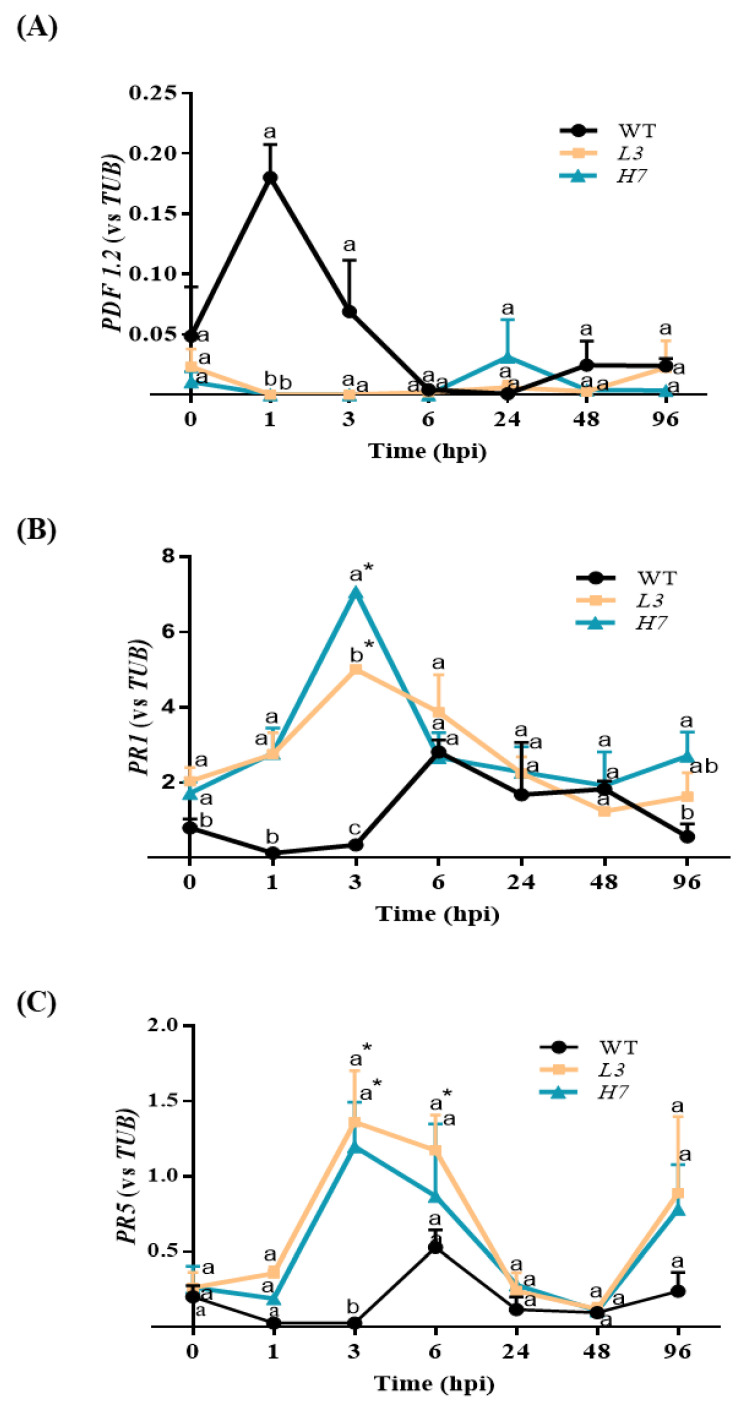
Defense responses in *Arabidopsis* seedlings after *F. oxysporum* infection. RT-qPCR analysis of *PDF1.2* (**A**), *PR1,* (**B**) and *PR5* (**C**) expression levels in WT, *L3,* and *H7 Arabidopsis* seedlings after *F. oxysporum* inoculation (0–96 hpi). Data represent the mean ± SEM of at least three independent experiments. Time point 0 hpi is the mean of the control values at the different time points. Different letters denote significant differences between the genotypes at each time point according to the Tukey´s multiple comparison test (*p* < 0.05). Asterisks denote significant differences with respect to the control (0 hpi) according to the Dunnett´s multiple comparison test (*p* < 0.05).

**Figure 7 antioxidants-12-01321-f007:**
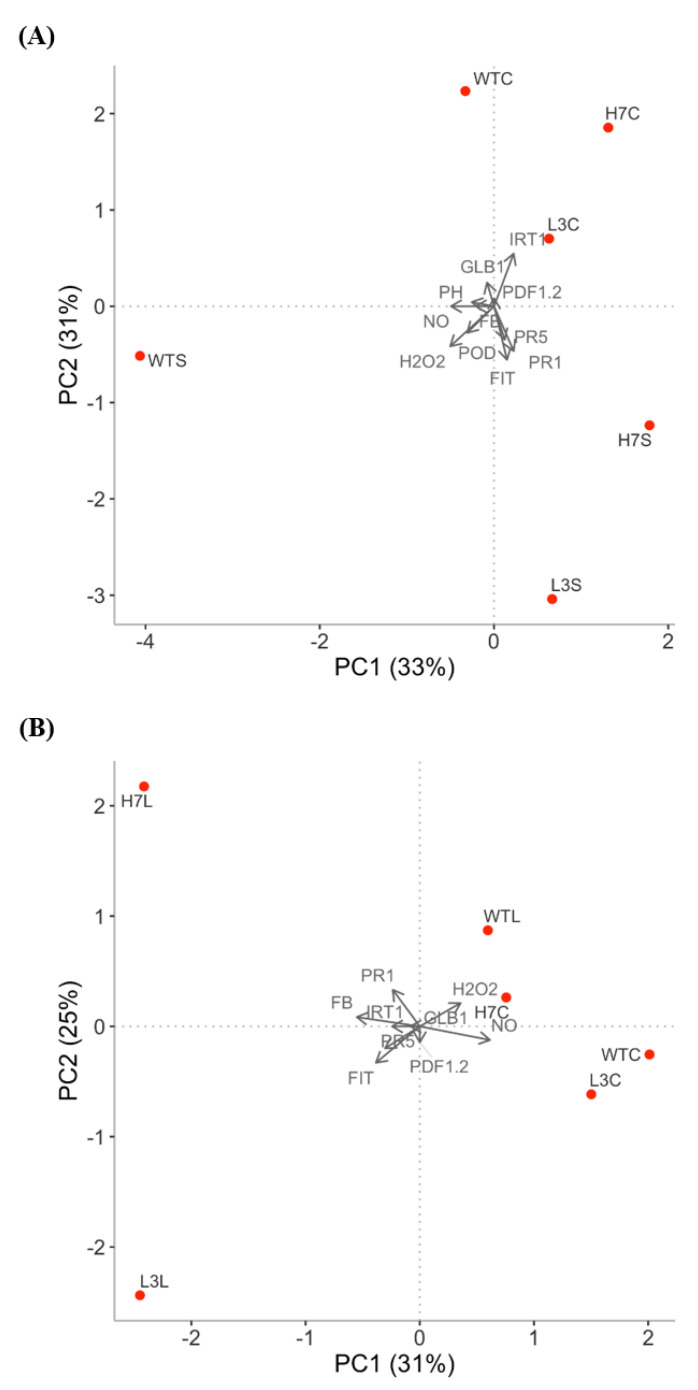
Biplot of *Arabidopsis* responses to *F. oxysporum***.** Short-time PCA (**A**) and long-time PCA (**B**) of WT, *L3,* and *H7* defense responses to *F. oxysporum*: fungal burden (FB), NO, and ROS (H_2_O_2_) production, Glb1 content (GLB1), defense response gene expression (PR1, PR5, and PDF 1.2), iron-related gene expression (IRT1 and FIT), phenol exudates (PH), and peroxidase activity (POD). The letters C, S, and L adjacent to the genotypes designate the control, short time, and long time, respectively. Each dot represents the mean value of the respective time points for each genotype, which is representative of at least three independent experiments. Principal component 1 (PC1, X-axis), X%, and principal component 2 (PC2, y axis), Y%, respectively.

## Data Availability

Should any raw data files be needed they are available from the corresponding author upon reasonable request.

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
