# Peer review of "Nitric Oxide and Globin Glb1 Regulate *Fusarium oxysporum* Infection of *Arabidopsis thaliana"

_antioxidants, 2023, doi:10.3390/antiox12071321_

Round 1
Reviewer 1 Report
In this study, Terron-Camero et al. investigate the interplay of the phytoglobin Gbl1 with NO production during infection of Arabidopsis thaliana with the fungal pathogen Fusarium oxysporum. They provide interesting and new insights into the plant defense responses in this context. In general, the authors provide rational and solid evidence by using a sufficient spectrum of techniques and genetic modifications.
From the experimental site a few additions should be made to complete the already compelling story (see major experimental points).

Overall the manuscript is excellently written and understandable. I could really follow the story and the interpretations of the data were accurate and reasonable. There is, however, a little work to do concerning spelling and phrasing in the text and in the figures (see major and minor points for the text).
Author Response
Referee(s)' Comments to Author:
Reviewer 1
In this study, Terron-Camero et al. investigate the interplay of the phytoglobin Gbl1 with NO production during infection of Arabidopsis thaliana with the fungal pathogen Fusarium oxysporum. They provide interesting and new insights into the plant defense responses in this context. In general, the authors provide rational and solid evidence by using a sufficient spectrum of techniques and genetic modifications. From the experimental site a few additions should be made to complete the already compelling story (see major experimental points).
Overall the manuscript is excellently written and understandable. I could really follow the story and the interpretations of the data were accurate and reasonable. There is, however, a little work to do concerning spelling and phrasing in the text and in the figures (see major and minor points for the text).
If the points mentioned in detail below can be addressed by the authors, this already fine article is ready for publication and will be a great contribution to the field.
- We thank reviewer 1 for their encouraging response.
Major experimental points:
- In general, the quality of the fluorescence images is not perfect but sufficient and the quantifications represent the representative images. Very nice. However, it would be very useful, if the brightness of the green fluorescence could be increased a little bit. Also bright field images are mandatory for Figures 1A and 5B. It would also be helpful to separate the time points with vertical dotted lines.
- We apologise for the lack of bright field images. We checked that the root is present and in focus before making the fluorescence images. We analysed an average of 10-12 roots per time point and per genotype, and use all the images with similar conditions for comparison purposes. Therefore, in one experiment, we imaged around 150-180 roots (10 roots x 5 time points x 3 genotypes). However, even though the roots were prepared serially, time was essential in order to avoid weakening the fluorescence, which explains why we did not obtain all bright field images.
As we did not want to modify the original images, the images do not look very bright, although the fluorescence is a realistic portrayal of the actual experiment. However, we have created a new figure, with an equal increase in contrast in all the images, and have decided to use these images in the paper.
- For all quantifications (Figure 1A, Figure 2B, C, Figure 4A, B, Figure 5B, Figure 6A, B, C and also the Supplementary Figures) it would help a lot, if you could use a colored scheme (for example WT in black, L3 in red and H7 in green). This would make the already clear data much easier to understand in a single view.
- Thank you for this suggestion. We have changed the initial grey colour and discontinuous line for the black and white images following your suggestion. However, we have used two colour-blind-friendly colours in order to make it easier to follow the results.
- In Figure 1B: Please add a t 0h to this dataset. Also a comparative analysis of the L3 and H7 mutants would be very interesting, similar to the dataset of Figure 1C.
-Data in this figure are expressed with respect to time point 0h, which explains its absence from the axis. The overexpressor lines showed high Glb1 expression, while the antisense lines exhibited some expression, partly due to the antisense sequence of these mutants. It is therefore difficult to distinguish the expression of the native gene from that of the antisense line. This explains why we performed the Western-blot, which shows the protein present in each genotype.
- In Figure 1C: Very nice controls of the knock-out mutant L3 and the over-expression mutant H7. Quantification (GLB1 levels divided by the Rubisco control bands, for examples via ImageJ) is needed here. Please do this for at least two independent Western blots (so basically and n2 or n3 of this experiment).
- We quantified Glb1 from two independent Western blots, which indicate the quantity of protein from infected roots as compared to control roots in supplementary Fig. S2, which may not have been described with sufficient clarity. We have now changed the quantification graph using the quantity of Glb1 protein divided by the Ponceau bands as suggested by this reviewer, which is now included in the new Supplementary Fig. S2
- In Figure 2A: Please add an early time point of your choice, where all plants are still healthy.
- We have added images from plants that were still alive at day 9.
- In Figure 2C: There is no need for the significance calculation relative to t 0 h in this graph. The main massage is that the levels between the mutants do not differ. So please write n.s., where needed.
-We have added n.s. as suggested.
- In Figure 3A: A t 0h and t 6h with the same conditions would be very interesting (also in addition and/or in comparison to Figure 1A).
-We firstly performed a time course analysis to detect peroxidase activity at 1, 3 and 24 h with the respective controls (C). We chose early time points (1 and 3 hpi) when NO begins to increase and at 24 h when NO was observed to peak. We only observed differences between infected and non-infected roots at 1 hpi, results which are included in Figure 3. However, we have added a new figure (Supplementary Figure S4) which shows peroxidase activity at 3 and 24 h and have described the results for greater clarity.
- In Figure 4C: Please also quantify the coloration here, as you nicely did in Figure 3A. Concerning the previous data sets of this Figure at t 48 h would be very helpful.
- We have now quantified coloration in this figure. As with peroxidase and phenol content, we performed a time course analysis of ferric chelate reductase activity, which, however, was only detected at 1 hpi. We have included the results for 3 and 24 h in a new figure (Supplementary Figure S4) and have also included a description of these results.
- In Figure 5A: If fluorescence was measured here, representative pictures (like in Figure 1A, for example) are needed. Also a small kinetic with t 0 h and t 6 h would be very interesting to broaden the finding.
- We measured root exudates using a plate multireader but did not image the roots. We have changed a sentence in Material and methods to avoid confusion and to provide a clearer explanation. We have added a 3 h time point to the graph when differences in phenol content between infected and non-infected root exudates were observed.
Major points for the text:
The introduction is exceptionally well-written. Very well done.
The Material & Methods- and the Results-section are very detailed and well written.
The part 3.3 is very nicely written. Also negative results, and their proper description and interpretation are necessary and useful to get the whole picture. I do rarely see it, but you did a good job here.
-We thank the reviewer for these useful comments
- The data sets depicting expression data in Figure 4A,B, Figure 6A,B,C and in the supplementary Figures should be more specific. You could write for example “IRT1 expression relative to TUB expression”.
- Following your suggestion, we have now added “relative to TUB expression” where necessary.
- In Results-Section 3.4 please add some examples of typical phenols produced in this context with citations.
- We have added some typical phenols produced in this context.
- Please do not refer to any data sets in your conclusion. Just make another additional section in the Results-section and just discuss your findings in your conclusion.
- We have added a new section to the results as suggested.
- In the Material & Methods- and the Results-section, the substance DCF is used to measure ROS. However, unfortunately, the claim that DCF is a resourceful substance to measure specifically H2O2 is completely outdated. It has an increased tendency to detect H2O2, but the side effects of this probe (especially the diffusible variant, which was used here) make this claim difficult. I do not demand to use another H2O2 specific probe, but all passages in the text (including the M&M and Results-Section, especially in section 3.4.) have to be rephrased. The only appropriate statement is that “total cellular ROS” were measured in this setting. This does not devaluate your findings at all. I want to stress that I worked for a long time on the ROS topic and this issue is important. For ascorbate it is the same issue. This substance is not a specific scavenger for H2O2. It can also scavenge radicals. So please change the text accordingly. For further reading concerning ROS measurements have a look at the recent guidelines:
- Following the comments made by this reviewer, we have changed the text accordingly.
Murphy, M.P., Bayir, H., Belousov, V. et al. Guidelines for measuring reactive oxygen species and oxidative damage in cells and in vivo. Nat Metab 4, 651–662 (2022). https://doi.org/10.1038/s42255-022-00591-z
Sies, H., Belousov, V.V., Chandel, N.S. et al. Defining roles of specific reactive oxygen species (ROS) in cell biology and physiology. Nat Rev Mol Cell Biol 23, 499–515 (2022). https://doi.org/10.1038/s41580-022-00456-z
Minor points for the text:
In line 1 of the Introduction: Please remove “during their lives”.
In line 11 of the Introduction: There must be a space between “amino” and “acids”.
In line 13 of the Introduction: Please write “during the interaction Arabidopsis and F. oxysporum”
In line 14 of the Introduction: Please write “described” instead of “evidenced”
In line 49 of the Introduction: An “m” is missing in “Programmed”
In line 55 of the Introduction: Please write “by using an genetic approach”
In line 3 of the Results: Please write “is still under investigation” instead of “is not well understood”
In line 8 of the Results: Please write “regulation during F. oxysporum infection of Arabidopsis plants”
- Following these useful comments, we have changed the text accordingly.
Reviewer 2 Report
A regulatory role of nitric oxide (NO) in the response of plants to pathogen infection attracts attention of plant physiologists for many years, but there are still some gaps to fill in. For example, phytoglobin (Glb) mediated fine mechanisms of particular plant-pathogen interactions are less studied. Therefore, the aim of presented work is clear. To achieve the aim, the authors used two Arabidopsis mutants contrasting in the synthesis of Glb and analyzed broad range parameters of their response to infection with Fusarium oxysporum. Data showed that both Glb1 antisense and overexpressor Arabidopsis mutants demonstrated a phenotype more resistant to pathogen infection than WT.
Although some interesting results were obtained in this work, the m/s suffers from several major issues. This is disappointing because this research group is known for high standard research and presentations.
1. Luck of methodological description, e.g. there is no mention of details of Western blot protocols, although blots were presented in the main m/s and Supplementary, though in Suppl. without any legends, a reader must guess. Another unclear method is assessment of survival (apart from description of the calculation method). What parameter was calculated? How were dead and alive cells (or tissues, organs, plants) determined?
2. Measuring many parameters, like NO, ROS, peroxidase activity etc, by staining the whole roots is possible but then quantification of the results should be accurate and requires clear description, just mentioning the program used is not enough. Presentation of all images together with graphs of quantification looks duplication to me, some of it could be in Supplementary.
3. Timing / time-coarse for measuring different parameters varies greatly, making it difficult to link them logically. For example, in Fig 3 peroxidase activity was measured 1 h after infection, while in Fig 5 phenols were measured 1 d after infection and H2O2 level starting with 6 h after infection. Extracellular phenols and H2O2 are substrates for apoplastic peroxidases but time of the measurements is not consistent here. Pathogen induced oxidative burst is broadly demonstrated to occur within minutes after infection with several follow ups. Burst of NO can also happen much earlier than 6 h after infection.
4. Text of the m/s requires serious tiding up by checking grammar and re-phrasing awkward pieces, e.g. the last sentence of the Abstract; in the Introduction – “As reactive molecule, NO levels should strongly be regulated”; in the Methods – “Phenolic compounds in the root exudates were quantified through UV light as described previously (365 nm) [55,56].”; in the Results Page 11 – “In our hands, oscillations in NO and H2O2 production appear to be inverted suggesting a regulation from one to each other, in a Glb1-dependent way.” (Meaning?); in Conclusions – “since fungal burden and other plant responses are similar in both mutant genotypes (Fig. 7)” (This is a full pre-last sentence!), and in the last sentence “...Glb1 may have a role in the regulation of certain genes in addition to regulating NO levels.” (Meaning is unclear)
Author Response
A regulatory role of nitric oxide (NO) in the response of plants to pathogen infection attracts attention of plant physiologists for many years, but there are still some gaps to fill in. For example, phytoglobin (Glb) mediated fine mechanisms of particular plant-pathogen interactions are less studied. Therefore, the aim of presented work is clear. To achieve the aim, the authors used two Arabidopsis mutants contrasting in the synthesis of Glb and analyzed broad range parameters of their response to infection with Fusarium oxysporum. Data showed that both Glb1 antisense and overexpressor Arabidopsis mutants demonstrated a phenotype more resistant to pathogen infection than WT.
Although some interesting results were obtained in this work, the m/s suffers from several major issues. This is disappointing because this research group is known for high standard research and presentations.
- Luck of methodological description, e.g. there is no mention of details of Western blot protocols, although blots were presented in the main m/s and Supplementary, though in Suppl. without any legends, a reader must guess. Another unclear method is assessment of survival (apart from description of the calculation method). What parameter was calculated? How were dead and alive cells (or tissues, organs, plants) determined?
- Based on the comments made by the reviewer, we have added new information that was lacking in the Material and methods section. We have also included legends for the supplementary figures in both the manuscript and under the supplementary figure for greater clarity.
- Measuring many parameters, like NO, ROS, peroxidase activity etc, by staining the whole roots is possible but then quantification of the results should be accurate and requires clear description, just mentioning the program used is not enough. Presentation of all images together with graphs of quantification looks duplication to me, some of it could be in Supplementary.
- We have added an accurate description of the quantification method and have maintained the reference which is also now described more clearly. In our view, the images would be easier to explain with the aid of the quantification graphs. However, we are willing to move the graphs to supplementary information if the editor so wishes.
- Timing / time-coarse for measuring different parameters varies greatly, making it difficult to link them logically. For example, in Fig 3 peroxidase activity was measured 1 h after infection, while in Fig 5 phenols were measured 1 d after infection and H2O2 level starting with 6 h after infection. Extracellular phenols and H2O2 are substrates for apoplastic peroxidases but time of the measurements is not consistent here. Pathogen induced oxidative burst is broadly demonstrated to occur within minutes after infection with several follow ups. Burst of NO can also happen much earlier than 6 h after infection.
-We performed time course analyses for all the parameters analysed at early time points (1 and 3 h) and a later time point which overlaps with the peak level of NO at 24 h; however, we have included in the manuscript only those time points that indicate significant differences. For greater clarity, we have now added some results not included in the previous manuscript to Supplementary Figure S4 and the new Fig. 5A. We now hope that the rationale for the time points used is clearer.
Although we observed a tendency for NO to increase in WT after 1 and 3h of infection, the differences detected were not always significant, which explains why we decided to use 6 h as the first time point for NO production.
Comments on the Quality of English Language
- Text of the m/s requires serious tiding up by checking grammar and re-phrasing awkward pieces, e.g. the last sentence of the Abstract; in the Introduction – “As reactive molecule, NO levels should strongly be regulated”; in the Methods – “Phenolic compounds in the root exudates were quantified through UV light as described previously (365 nm) [55,56].”; in the Results Page 11 – “In our hands, oscillations in NO and H2O2 production appear to be inverted suggesting a regulation from one to each other, in a Glb1-dependent way.” (Meaning?); in Conclusions – “since fungal burden and other plant responses are similar in both mutant genotypes (Fig. 7)” (This is a full pre-last sentence!), and in the last sentence “...Glb1 may have a role in the regulation of certain genes in addition to regulating NO levels.” (Meaning is unclear)
- A native English speaker has thoroughly revised the manuscript to avoid any confusion and for greater clarity.
Round 2
Reviewer 1 Report
The autors fully adressed my issues and suggestions. I really like the new color scheme of the graphs. The results are much clearer now. A very nice article. This is ready for puplication. Good work.
Reviewer 2 Report
The change authors made in the revision helped to improve m/s. Response to my comments and corresponding changes in the main text are satisfactory.
Minor typos are still present but in general English is better.